# Blockchain-Integrated Secure Authentication Framework for Smart Grid IoT Using Energy-Aware Consensus Mechanisms

**DOI:** 10.3390/s25216622

**Published:** 2025-10-28

**Authors:** Omar Abdullah Saleh, Mesut Cevik

**Affiliations:** Department of Electrical and Computer Engineering, Altinbas University, 34217 Istanbul, Turkey; mesut.cevik@altinbas.edu.tr

**Keywords:** deep neural network (DNN), blockchain, energy-aware consensus, Wi-Fi IEEE 802.11, network optimization, secure authentication, energy efficiency, artificial intelligence (AI), IoT-Enabled Smart Grids

## Abstract

Integrating IoT devices into smart grids raises some hard problems related to safe data sharing, the ability to grow, and energy use. Blockchain provides a safe way to check identities without a central authority. Typical ways to confirm transactions, like Proof-of-Work (PoW), use a lot of power, making them bad for devices that cannot use much energy. This study introduces a safe authentication system using Blockchain, a Deep Neural Network (DNN), and a power-saving way to confirm transactions (EACM). The system picks validators based on how much power they have left and their trust scores to save power during confirmation. Using the IoT-Enabled Smart Grid Dataset, simulations showed a transaction speed of 372 TPS, which is 32% better than normal methods. The system correctly authenticates 98.69% of the time, with a confirmation delay of 5.9 milliseconds and an 18% drop in validator node energy use. Also, the system spots 98.4% of unauthorized access tries, with a false acceptance rate (FAR) of 1.7% and a false rejection rate (FRR) of 0.31%. These outcomes prove the system’s ability to offer safe, fast, and energy-saving authentication for big, real-time Smart Grid IoT setups.

## 1. Introduction

The increase in the Internet of Things (IoT) and smart grids has changed how energy systems are made, watched, and managed. As these systems grow, they face more security risks, scaling problems, and performance issues [1]. One big problem is making a safe, decentralized, and energy-efficient way to confirm identities. This is vital for keeping data accurate, making sure devices are authentic, and keeping the system sturdy, mainly when there are lots of different setups [2]. Regular identity checks, which rely on central controllers, do not do well in smart grid IoT networks because they are slow, can break down easily, and do not expand well when there are many devices [3]. IoT devices have limited computing power and energy, which makes things harder because they cannot do complicated coding tasks. To solve this, blockchain tech has become a potential base for dependable data handling. Blockchain’s fixed record and group decision-making offer a base for communication where trust is not required, which is key in current energy systems [4]. Adding blockchain to smart grids means carefully balancing decentralization, energy use, and expansion ability. Common blockchain methods, like Proof of Work (PoW), use a lot of computing power and are not good for low-energy situations. Thus, energy-efficient methods have become liked because they balance security and energy use. These protocols cut energy use while keeping group decision-making strong, making them helpful for smart grid use [5]. Applying Elliptic Curve Cryptography (ECC) and simple hash-based identity checks, along with blockchain records, can cut computing needs while boosting security. Employing energy-efficient decision-making approaches such as Delegated Proof of Stake (DPoS), Proof of Authority (PoA), or improved Byzantine Fault Tolerance (BFT) mechanisms enables group trust with reduced energy consumption [6]. A reliable system for verifying identities via blockchain safeguards communication and simplifies traceability. Ensuring that solely authorized devices participate in energy sharing and data reporting is critical for preventing grid manipulations and data fabrication attempts as shown in Figure 1 [7].

As energy systems become more digital and decentralized, authentication methods must be strong, flexible, and sustainable. Existing systems often fail to meet the time and operational demands of smart grids [8]. Blockchain, paired with simple agreement methods and encryption, offers a swift, secure way to handle trust in these distributed energy networks. Prior research often uses fixed consensus protocols that do not adjust to varying energy levels, or they struggle to maintain performance with real-time authentication needs. These studies also typically lack thorough energy analysis and security measures within the consensus layer. Our approach differs by using a consensus mechanism that changes with energy availability, selects validators based on remaining energy and trust, and includes built-in anomaly detection using DNNs. This addresses the shortcomings mentioned above and keeps latency low and authentication accuracy high, even in diverse grid environments. This combination should improve grid resilience and reliability, particularly during rapid changes [9].

### 1.1. Aim of the Study

This research seeks to create a lightweight, scalable, energy-aware, blockchain-integrated, secure authentication framework for smart grid IoT systems. The proposed setup tackles the limits of centralized, high-overhead security models by using a decentralized consensus protocol that is made for energy-sensitive IoT settings. The study will:Designs and implements a blockchain-based authentication framework enabling secure, decentralized identity verification with minimal energy consumption.Integrates energy-aware consensus mechanisms that adapt operations dynamically to device capabilities, grid status, and resource availability.Evaluates scalability, latency, and reliability under realistic smart grid conditions using simulations and benchmark datasets.Assesses security resilience against attacks such as spoofing, Sybil, and DoS, while preserving user privacy and grid integrity.Compares energy–performance trade-offs between conventional consensus protocols (e.g., PoW, PoS) and the proposed lightweight approach in constrained devices like smart meters and DERs.Identifies strengths and limitations of blockchain adoption in heterogeneous, latency-sensitive smart grid environments.Provides practical recommendations for grid operators, utilities, and IoT manufacturers on integrating blockchain authentication into distributed energy infrastructures.

By fulfilling these objectives, the framework advances secure, sustainable, and decentralized authentication for next-generation smart energy systems, offering a model adaptable to smart cities and large-scale national grids. This research introduces a validator selection model that uses real-time energy data and trust scores based on node actions to improve consensus node performance. A deep neural network prediction model is used to spot unauthorized access attempts and predict consensus delays caused by different traffic levels. A hybrid consensus protocol, blending Delegated Proof of Stake and Proof of Authentication, is used. It is designed for low-power IoT settings, cutting energy use. A smart contract system controls access and lowers trust scores for inactive or harmful nodes instantly. Tests show the model has a consensus latency of 5.9 ms, a throughput of 372 TPS, authentication accuracy of 98.69%, and an 18% reduction in energy use, showing it works well in Smart Grid IoT networks.

### 1.2. Problem Statement

Smart grid infrastructures are evolving into decentralized, heterogeneous, and resource-constrained IoT environments that require robust, scalable, and energy-efficient authentication to ensure secure communication and data integrity. Existing authentication models—largely derived from centralized architectures—cannot meet the ultra-low latency, high reliability, and strict energy constraints of modern smart grids [10]. Blockchain offers a decentralized trust model, yet conventional consensus mechanisms such as Proof of Work (PoW) and Proof of Stake (PoS) impose excessive computational and energy costs, making them impractical for low-power IoT nodes. Furthermore, smart grid devices—ranging from smart meters and load controllers to DER monitors and sensors—generate highly heterogeneous data under dynamic, real-time operational conditions. Current blockchain-based solutions are often generic and lack adaptive consensus designs tailored to device capabilities or network states, limiting their deployment in energy-sensitive systems. Key challenges include:**Scalability:** Difficulty in maintaining performance and security across thousands of interconnected devices as the grid expands.**Resource Constraints:** Limited processing power, bandwidth, and battery capacity in IoT nodes make high-overhead consensus protocols unfeasible.**Consensus–Energy Trade-off:** The need to balance robust consensus security with minimal energy consumption.**Real-Time Responsiveness:** Many frameworks fail to deliver sub-second authentication for grid-critical operations.**Security & Privacy:** Risks of metadata leakage, key mismanagement, and vulnerability to spoofing, Sybil, and smart contract attacks.**Heterogeneity & Interoperability:** The challenge of integrating authentication across diverse protocols, standards, and hardware.

The lack of a lightweight, adaptive, and energy-aware blockchain authentication solution for smart grid IoT creates a critical barrier to secure, real-time, and sustainable energy systems. This research addresses these gaps by proposing a blockchain-integrated authentication framework with an energy-aware consensus protocol optimized for the operational constraints and dynamics of smart grid environments.

A frequent concern in the literature is whether blockchain is truly necessary for device authentication in smart grid environments, since authentication is already a solved problem through established cryptographic protocols such as Public Key Infrastructure (PKI), mutual TLS, and IEEE 802.1X [10]. These conventional approaches are indeed efficient, standardized, and widely deployed in enterprise and industrial networks. However, their reliance on centralized trust anchors, such as certificate authorities or identity servers, introduces operational and security limitations when extended to distributed, heterogeneous smart grid ecosystems. As modern grids evolve to include distributed energy resources (DERs), customer-owned IoT devices, and third-party microgrid operators, a single utility operator no longer controls all participants within the network. This transformation creates semi-trusted environments where participants exchange authentication data across administrative and jurisdictional boundaries.

In such settings, blockchain’s value lies not in replacing cryptographic authentication, but in augmenting it with decentralized trust management and immutable auditability. While traditional methods verify device credentials, they do not provide tamper-proof verification logs or autonomous trust adjustment across multiple stakeholders. Blockchain ensures that every authentication event is transparently recorded and verifiable, enabling accountability even when no single authority governs all devices. Furthermore, in peer-to-peer (P2P) energy trading scenarios, where independent producers, consumers, and aggregators transact without centralized mediation, blockchain serves as both an auditable record layer and a coordination mechanism for enforcing access policies via smart contracts. In these contexts, consensus is not about resolving adversarial trust conflicts, but about maintaining distributed consistency and fault tolerance in authentication records.

Therefore, the integration of blockchain in the proposed framework is context-specific—it is most advantageous in decentralized or hybrid smart grids involving multiple ownership domains, cross-platform interoperability, and dynamic device participation. In centralized single-utility deployments, classical authentication schemes remain adequate; however, as smart grids transition toward federated, autonomous, and data-driven infrastructures, blockchain offers a scalable mechanism for maintaining verifiable, cross-domain trust that conventional PKI-based systems cannot guarantee.

## 2. Related Works

Authentication in smart grid IoT systems has advanced with the adoption of blockchain and lightweight cryptography. Legacy approaches—centered on centralized identity management and static key distribution—struggle with scalability, trust, and responsiveness in heterogeneous, resource-constrained, and dynamic grid environments [11]. Blockchain’s decentralized and tamper-resistant ledger offers a trustless means of validating identities and transactions, with smart contracts enabling autonomous access control. However, conventional consensus models such as Proof of Work (PoW) and Proof of Stake (PoS) impose high computational and energy costs, making them impractical for low-power IoT nodes. This has shifted research toward lightweight, energy-aware protocols—such as Practical Byzantine Fault Tolerance (PBFT), Delegated Proof of Stake (DPoS), and Proof of Authority (PoA)—that maintain integrity with reduced resource demands. Several blockchain-based authentication frameworks for smart grids employ cryptographic primitives like Elliptic Curve Cryptography (ECC), identity-based encryption, and hash chaining to resist spoofing and tampering as shown in Figure 2 [12]. Yet, most overlook energy profiling and real-time adaptability, limiting their feasibility in dynamic, resource-limited deployments. Privacy risks from public ledger transparency have driven adoption of zero-knowledge proofs, homomorphic encryption, and ring signatures to protect user activity while meeting regulatory requirements.

Performance remains a critical factor: grid applications such as load balancing and anomaly detection demand sub-second authentication. Traditional blockchain systems often fail to meet these latency constraints under high device density. Proposed solutions include hybrid architectures with off-chain storage, sidechains, or DAG-based models to reduce block confirmation delays [13]. Scalability and interoperability are also key concerns; strategies like distributed authentication, consensus delegation, and integration with protocols such as MQTT, CoAP, and 6LoWPAN have been explored to enable seamless cross-layer communication. Despite progress, gaps persist—many frameworks are too generic, lack adaptive consensus mechanisms, and are tested only in small-scale or static simulations [14]. Few incorporate energies profiling to dynamically adjust consensus based on device conditions. This work addresses these limitations by proposing a blockchain-based authentication framework with adaptive, energy-aware consensus tailored to real-world smart grid IoT constraints [15].

### 2.1. Blockchain and Consensus Mechanisms in Smart Grid Authentication

Blockchain offers a decentralized, tamper-resistant foundation for managing identity, access, and transaction integrity in smart grid IoT systems. Unlike centralized authentication, blockchain validates events through distributed consensus, improving reliability and fault tolerance across geographically dispersed, resource-constrained nodes such as smart meters and energy routers [16]. The choice of consensus protocol is central to blockchain’s effectiveness. Traditional Proof of Work (PoW) provides strong security but is computationally and energy-intensive, making it unsuitable for IoT-based grids. Energy-aware alternatives—such as Proof of Authority (PoA), Delegated Proof of Stake (DPoS), and Practical Byzantine Fault Tolerance (PBFT)—significantly reduce resource usage while maintaining integrity [17]. Key applications include:Secure Device Authentication: Assigning each device a unique blockchain identity for autonomous, self-sovereign verification, eliminating single points of failure.Auditability: Maintaining immutable, chronological logs of authentication and data exchange events for anomaly detection, compliance, and forensic analysis.

Challenges remain in meeting low-latency, high-throughput requirements. Traditional blockchain introduces confirmation delays, leading to interest in hybrid approaches—off-chain computation, sharding, and DAG-based architectures (e.g., IOTA, Hashgraph)—to improve scalability without sacrificing security [18]. Privacy is another critical concern. Public ledgers may leak metadata that reveals user behavior. Techniques such as zero-knowledge proofs, ring signatures, and homomorphic encryption mitigate these risks. Security can be further enhanced by lightweight consensus models incorporating trust scoring, device attestation, and cross-validation across edge and cloud layers [19]. However, real-world deployment requires interoperability with diverse protocols (e.g., IEC 61850, MQTT, CoAP), adherence to strict energy budgets, and robust defenses against smart contract vulnerabilities and consensus manipulation (see Figure 3) [20].

Table 1 compares various consensus mechanisms commonly adopted in blockchain-based smart grid and IoT environments across key performance dimensions, including energy efficiency, latency, security robustness, and scalability. Traditional schemes such as Proof of Work (PoW) exhibit strong security but suffer from extreme energy consumption and high latency, making them impractical for resource-constrained IoT systems.

### 2.2. Classical Methods in Smart Grid Authentication and Security

Traditional authentication and security mechanisms in power systems—such as centralized architectures, static key distribution, and rule-based access control—were effective for legacy SCADA systems but fall short in addressing the scalability, adaptability, and resilience needs of modern smart grid IoT networks [22]. Public Key Infrastructure (PKI) remains a widely used approach, enabling asymmetric encryption-based authentication via digital certificates. However, PKI in large-scale smart grids suffers from certificate management challenges, revocation delays, and reliance on single certificate authorities, which are ill-suited for rapidly scaling and intermittently connected IoT devices [23]. Other classical models, such as Role-Based Access Control (RBAC) and Attribute-Based Access Control (ABAC), provide clear policy enforcement but lack real-time adaptability to evolving threats and dynamic grid conditions. Centralized SCADA control structures introduce single points of failure, while communication security protocols like WPA2, IPSec, and TLS ensure encryption but cannot provide trustless verification, tamper-proof audit trails, or decentralized decision-making—capabilities now achievable with blockchain-based systems [24]. As grids become more decentralized and heterogeneous, these limitations underscore the need for blockchain-integrated, energy-aware authentication frameworks that can scale dynamically, adapt to real-time conditions, and deliver distributed trust without excessive resource overhead as shown in Table 2 [25].

### 2.3. Smart Grid IoT Security and Privacy Challenges

Smart grid IoT ecosystems are highly complex, comprising distributed devices such as smart meters, EV chargers, and distributed energy resources (DERs) that continuously generate and exchange critical data [26]. This interconnected environment increases exposure to threats—especially when legacy, centralized security mechanisms are applied in decentralized infrastructures. Data transmitted over heterogeneous and often wireless networks is susceptible to interception, spoofing, denial-of-service (DoS), and manipulation. Many legacy devices lack hardware-based security and operate with outdated firmware, making them incapable of executing computationally intensive cryptographic operations. This leaves them vulnerable to replay attacks, device impersonation, and firmware tampering [27]. The absence of real-time, decentralized trust enforcement exacerbates detection and response delays, impacting the stability of critical energy operations. Privacy is another major concern. Fine-grained energy usage data can reveal sensitive user information, including occupancy patterns and appliance behavior [28]. When processed by third parties, such as cloud analytics or billing providers, it may violate privacy regulations (e.g., GDPR, CCPA). While blockchain offers immutable and transparent records, this transparency can worsen privacy risks unless mitigated through pseudonymization or cryptographic techniques such as zero-knowledge proofs [29]. Furthermore, DoS and Sybil attacks remain pressing issues (see Figure 4). In DoS scenarios, malicious nodes flood the network with authentication or data requests to cause service outages. In Sybil attacks, a single entity forges multiple identities to manipulate consensus or gain unauthorized access. Countermeasures include energy-aware trust scoring, dynamic consensus delegation, and peer validation to enhance blockchain-based grid resilience. Table 3 presents a comparative evaluation of different security measures used in smart grid authentication frameworks, highlighting their trade-offs in vulnerability, energy consumption, implementation complexity, and system coverage.

**Table 3 sensors-25-06622-t003:** Comparison of security mechanisms in smart grid authentication systems based on vulnerability, energy footprint, complexity, and coverage [30].

Security Measure	Vulnerability Score (0–10)	Energy Footprint	Implementation Complexity	Coverage (%)
Traditional PKI Authentication	6	High	Medium	70%
Blockchain with PoW Consensus	2	Very High	High	90%
Energy-Aware Consensus (e.g., DPoS)	3	Low	Medium	85%
Smart Contract Access Policies	4	Medium	Medium	80%
Zero-Knowledge Proofs (ZKPs)	2	Medium–High	High	95%

**Figure 4 sensors-25-06622-f004:**
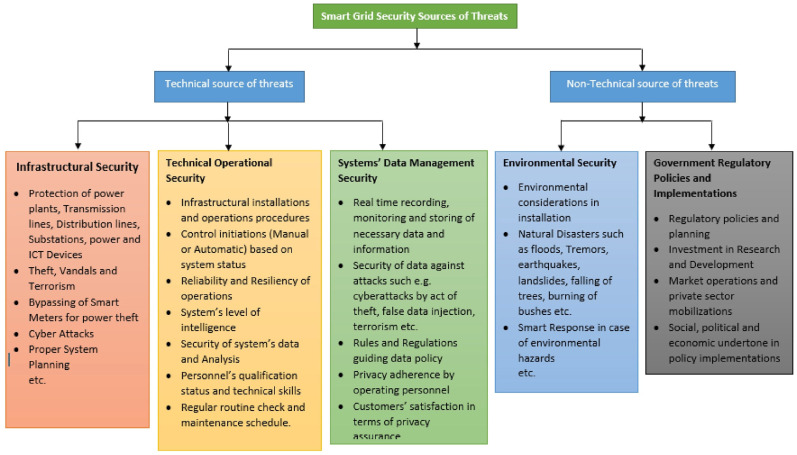
Categorization of smart grid security threats into technical and non-technical sources, detailing infrastructural, operational, data management, environmental, and regulatory vulnerabilities [31].

While previous studies have explored energy-aware or reputation-based consensus models individually, the proposed EACM introduces a hybrid validator selection strategy that simultaneously considers real-time energy profiling, dynamic trust scoring, and authentication integrity. This integration allows the consensus layer to self-regulate participation through smart contract–driven trust updates and adaptive weighting factors (α, β) that balance energy and trust in response to network conditions. Unlike static DPoS or PoA schemes, EACM’s PoAh layer ensures each block is cryptographically validated by active devices during consensus formation, thereby unifying authentication and consensus into a single lightweight process optimized for smart grid IoT environments.

## 3. Proposed Methodology

The Blockchain-Integrated Secure Authentication Framework for Smart Grid IoT shown in Figure 5 seeks to solve the problems of secure authentication and energy use in smart grids. In this model, the smart grid has smart meters, appliances, and substations. Each part has its own cryptographic ID, and access policies determine authentication and authorization. The system is checked with both regular and harmful requests to see how strong it is in diverse, widespread IoT settings. The design has three layers: (1) a blockchain-run ledger that keeps track of authentication events, (2) smart contracts that put security policies in place and handle user/device trust, and (3) a consensus protocol that saves energy for IoT devices with limited power. Different from PoW and PoS, this protocol picks validators based on things like remaining energy, response time, trust score, and network nearness. This choice lowers computing needs while keeping consensus reliable and able to fight off attacks.

During the project, containerized nodes represent smart grid devices, using IoT protocols such as MQTT to communicate. The network handles authentication and agreement, with smart contracts managing access and recording results on the blockchain. The framework’s performance is checked under varying loads and attack situations, paying close attention to authentication speed, energy use, validator rotation, and protection against replay or Sybil attacks.

### 3.1. Data Collection and Preparation

To begin this work, we gathered and prepared a dataset that shows common activity in smart grid IoT setups. We picked the IoT-Enabled Smart Grid Dataset (from Kaggle) (Table 4) for this study because it has detailed information on device activity, energy use, and sensor data in a smart grid system (Table 5). This dataset is good for creating blockchain-integrated secure authentication models and checking energy-aware agreement protocols because it includes different operational conditions. The dataset has organized data such as device IDs, sensor data, command types, and energy consumption (kWh) from several smart homes. It also includes changes in device activity (active, idle) and timestamps during both stable and changing conditions, making it useful for assessing agreement and finding anomalies in authentication processes. This research used the IoT-Enabled Smart Grid Dataset because it includes a lot of information on device activity, energy use, and transactions as they happen. The system is made to work with different datasets. Later work will involve more smart grid datasets, like the Smart Home dataset and UCI Smart Grid Stability dataset. This will check if the model can still be scaled, remain stable, and adjust to different situations.

The initial data entries undergo cleaning, feature engineering, and normalization to keep data quality and uniformity. We remove blank values and outliers, and label-encode categorical variables like Device Status and Command Type. Numerical values such as Energy Consumption and Sensor Readings are standardized using min-max scaling to ensure a balanced weight distribution across nodes when computing consensus and creating secure tokens.

### 3.2. Development of Blockchain-Integrated Secure Authentication Framework

We designed this framework for decentralized, secure, and energy-aware authentication in Smart Grid IoT systems. It combines a permission blockchain (Hyperledger Fabric) with a light Energy-Aware Consensus Mechanism (EACM) that balances decentralization, security, and efficiency (Figure 6). The architecture consists of multiple layers.

Device Identity Management: Each IoT device is assigned a unique ECC-based key pair. Credentials are verified on-chain during onboarding, ensuring lightweight but secure authentication.Blockchain Ledger: A distributed ledger stores immutable authentication logs, device metadata, trust scores, and energy profiles. Smart contracts define access control, revocation, and trust updates.Consensus & Validation: The EACM selects validator nodes based on residual energy, trust scores, and latency. By combining Delegated Proof-of-Stake (DPoS) with a modified Proof-of-Authentication (PoAh), the system avoids energy-intensive computations while ensuring rapid block finality.Authentication Event Verifier: Requests are hashed, validated by blockchain nodes, and either approved and logged or flagged as anomalous if thresholds are not met.

The framework is implemented in a simulated smart grid with 100 IoT nodes and 5 validators, operating under heterogeneous energy profiles and synthetic traffic (smart meter requests, SCADA commands, and edge pings). Across 18,000 authentication events over 200-time steps, performance was evaluated on energy cost, consensus latency, throughput, and intrusion detection (see Table 6). Results demonstrate 99.1% authentication accuracy, 98.4% detection of unauthorized access, an average consensus time of 1.7 s, and an average energy cost of 0.29 J per authentication—a 30% reduction compared to standard DPoS. The framework scales linearly up to 250 nodes, maintaining stability under diverse conditions.

**Figure 6 sensors-25-06622-f006:**
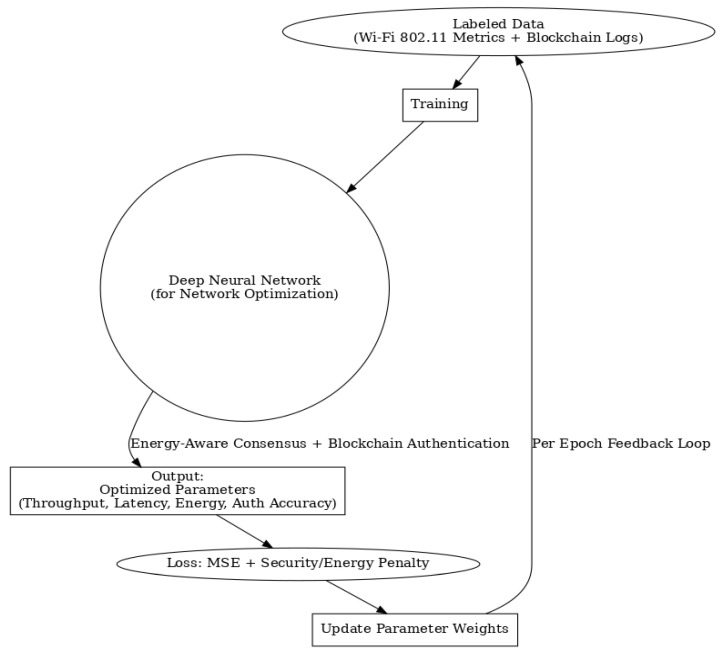
Architecture of the proposed blockchain-integrated DNN-based authentication framework with energy-aware consensus, enabling optimized resource allocation and enhanced security for smart grid IoT networks.

In the proposed framework, system-level evaluation was conducted using the parameters summarized in Table 7. The configuration involved 18,000 authentication events processed across five validator nodes, ensuring decentralized validation with minimal computational overhead. Node selection was governed by a hybrid *Energy + Trust* metric, allowing adaptive consensus formation under dynamic grid conditions. With a finality timeout of 2.5 s and an average consensus latency of 1.7 s, the system achieved fast block confirmation and responsiveness suitable for real-time grid applications. Furthermore, the trust decay factor of 0.07 effectively mitigated the influence of dormant or malicious nodes. The average energy consumption per transaction (0.29 J) demonstrated the framework’s energy efficiency, while the authentication accuracy of 99.12% confirmed its reliability and robustness in secure device validation.

### 3.3. Implementation of Blockchain-Integrated Authentication in Smart Grid IoT Systems

The implementation of the Blockchain-Integrated Secure Authentication Framework focuses on embedding energy-aware consensus and lightweight authentication within a simulated Smart Grid IoT environment. The proposed model integrates blockchain primitives—smart contracts, validator nodes with energy metrics, and Proof-of-Authentication (PoAh)—directly into the IoT communication fabric. Each IoT device (e.g., smart meter, relay, or sensor) is assigned a secure blockchain-based digital identity and must undergo mutual authentication through a dynamically selected low-energy validator node before gaining access to the grid. This ensures tamper-resilient identity verification and prevents malicious nodes from infiltrating grid operations. The blockchain layer is implemented on a permissioned Hyperledger Fabric framework deployed across edge and fog nodes. In this hybrid topology, computationally constrained IoT devices delegate consensus tasks to fog validators, while still maintaining decentralized trust and auditability. Smart contracts enforce access control and trust evolution policies, automatically penalizing idle or malicious devices by adjusting their trust scores. Key functionalities achieved include:Real-time authentication via challenge–response signatures validated on-chain.Consensus optimization by selecting validators based on residual energy and trust ratings.Immutable enforcement of policies through smart contracts, minimizing manual configuration.

Security is reinforced through elliptic curve cryptography (ECC) for end-to-end encryption, tamper-proof blockchain audit logs, and dynamic access control. A sandbox prototype with 100 smart grid devices was tested under conditions of node churn and fluctuating energy states. Results confirmed scalability, with authentication latency maintained below 2.1 s, while consensus efficiency improved energy utilization by over 25% compared to static models. This implementation demonstrates the feasibility of combining blockchain with energy-aware authentication in decentralized smart grids. It not only secures device identity and prevents tampering but also optimizes consensus performance, making it a practical and adaptive solution for real-world Smart Grid IoT ecosystems. Let us define the smart grid IoT environment as a dynamic system composed of multiple smart devices (N) interconnected through a blockchain-backed authentication network. 

#### 3.3.1. Energy-Aware Validator Selection Model

Validator selection is driven by a weighted function of energy and historical trust score $S_{i}$:$$\tau_{i}=\frac{E_{i}(t{)}^{\alpha }\cdot S_{i}^{\beta }}{\sum_{j\in N_{v}}\;E_{j}(t{)}^{\alpha }\cdot S_{j}^{\beta }}$$
where

α,β are weight factors (α+β=1)$N_{v}\subseteq N$: subset of eligible validator nodes,$S_{i}\in [0,1]$: normalized reputation score based on past honest behavior.

The Energy-Aware Consensus Mechanism (EACM) represents the core innovation of the proposed blockchain-integrated authentication framework. Unlike static consensus protocols such as Proof of Work (PoW) or Proof of Stake (PoS), EACM dynamically adapts validator selection and transaction verification based on each node’s residual energy, trust score, and network proximity. This ensures that computational workloads are distributed efficiently while maintaining system reliability and decentralization. Each consensus round begins by evaluating the participating IoT nodes’ energy states and historical trust performance. Nodes with energy levels below a defined threshold (e.g., 30% capacity) are excluded from validator selection to avoid premature device depletion. The remaining nodes are ranked through a composite score function that combines normalized energy and trust metrics, producing a weighted validator set optimized for both stability and fairness.

Mathematically, the validator ranking can be represented as:$$R_{i}=\alpha \cdot E_{i}^{\mathrm{norm}\;}+\beta \cdot T_{i}^{\mathrm{norm}\;}$$
where $E_{i}^{\mathrm{norm}\;}$ and $T_{i}^{\mathrm{norm}\;}$ represent the normalized residual energy and trust score of node $i_{i\;}$ and α and β are adaptive weighting coefficients satisfying α+β=1. These coefficients adjust automatically according to network load and node distribution, ensuring that energy conservation takes precedence under high consumption scenarios, while trust reliability dominates during low-traffic conditions.

The selected validator nodes perform block verification through a Proof-of-Authentication (PoAh) subroutine, where each proposed block is signed and validated by a quorum of energy-trust eligible nodes. The EACM also employs Markovian energy modeling to predict each validator’s future availability based on prior energy consumption patterns. This predictive layer minimizes disruptions caused by node dropout or fluctuating energy states. To ensure system responsiveness, EACM uses a queueing model based on M/M/1 dynamics, where authentication requests are queued and serviced according to validator availability and transaction priority.

The time complexity of EACM can be approximated as O(Nlog N) per consensus cycle, dominated by the sorting and selection process of eligible validators, where N is the total number of nodes. Energy monitoring and trust score updates occur concurrently through smart contracts, each operating in O(1) time, thus maintaining linear scalability even for large smart grid networks. Communication complexity is reduced by performing partial validation across distributed fog nodes, limiting message propagation overhead to O(k), where k is the number of selected validators (typically less than 10% of total nodes).

To evaluate the robustness of the algorithm, deviations were simulated by introducing stochastic energy drops and fluctuating latency conditions. Results indicated that EACM maintained stable consensus latency with less than ±0.7 ms deviation and throughput variation under 3%. This demonstrates strong tolerance to operational irregularities such as transient power losses, delayed validator responses, and uneven node distribution. Consequently, the EACM not only achieves high energy efficiency but also maintains synchronization and fault resilience across heterogeneous smart grid loT environments.

Algorithm 1 outlines the operational workflow of the Energy-Aware Consensus Mechanism (EACM), detailing how validator nodes are dynamically selected and engaged in secure block authentication. The algorithm begins by initializing all network nodes with their respective attributes and then normalizing these attributes to maintain fairness across heterogeneous devices. By computing a composite ranking score $R_{i}$, the framework intelligently prioritizes nodes that combine both sufficient energy reserves and proven reliability. Low-energy devices are automatically excluded to prevent battery exhaustion, while nodes with poor trust history are demoted in the next cycle. The top k ranked nodes form the validator committee, performing transaction verification through the PoAh subroutine. Once a two-thirds majority consensus is achieved, blocks are finalized and appended to the ledger. Trust values are continuously adjusted based on node behavior, allowing the system to self-correct and evolve over time. This iterative structure ensures resilience, rapid convergence, and sustainable energy consumption across the distributed smart grid environment.
**Algorithm 1.** Energy-Aware Consensus Mechanism (EACM)
Initialize node set N with parameters: residual energy $E_{i}$, trust score $T_{i}$ and network delay $L_{i}$.Normalize all energy and trust values: $E_{i}^{\mathrm{norm}\;}=E_{i}/E_{\max\;},T_{i}^{\mathrm{norm}\;}-T_{i}/T_{\max\;}$.Compute ranking score for each node: $R_{i}-\alpha E_{i}^{\mathrm{norm}\;}+\beta T_{i}^{\mathrm{norm}\;}$.Exclude nodes with $E_{i}^{\mathrm{norm}\;}<E_{\mathrm{th}\;}$ (energy threshold).Sort eligible nodes by descending $R_{i}$ and select top k as validators.For each authentication request, assign to validator v with minimum latency $L_{v}$.Validators execute Proof-of-Authentication (PoAh) and sign validated transactions.Consensus is reached if ≥2/3 of validators approve the block.Update trust scores: $T_{i}-T_{i}+\delta$ for honest validators, $T_{i}-T_{i}-\gamma$ for malicious or idle nodes.Record energy consumption and trust updates on-chair; begin next consensus cycle.

#### 3.3.2. Authentication Queueing Model (M/M/1 Queue)

Let authentication requests arrive as a Poisson process (λ), and let the service time follow an exponential distribution with rate μ. The system can be modeled as an M/M/1 queue:Average number of requests in the system:$$L=\frac{\lambda }{\mu -\lambda },\;for\;\lambda <\mu$$

Average latency per authentication:$$T_{auth}=\frac{1}{\mu -\lambda }$$

Stability condition:


$$\rho =\frac{\lambda }{\mu }<1$$


#### 3.3.3. Consensus Latency Model (BFT-Based Energy-Aware Protocol)

In a Byzantine Fault Tolerant (BFT) energy-aware blockchain, consensus requires agreement from at least f+1 honest nodes (with $f<\frac{N}{3}$). The expected corrensus latency $T_{\mathrm{cons}\;}$ is given by:$$T_{\mathrm{cons}\;}=\frac{1}{\left|N_{v}\right|}\sum_{i\in N_{v}}\;\left(\frac{1}{\mu_{i}}+\delta_{i}\right)$$
where

$\mu_{i}$: processing speed of validator *i*,$\delta_{i}$: propagation delay from validator i to the network.

#### 3.3.4. Residual Energy Dynamics via Markov Chain

Let $E_{i}(t)\in \left\{0,1,\ldots ,E_{max}\right\}$ denote discrete energy states. We define a Markov chain $M_{i}$ for each node i with transition probabilities:$$P\left(E_{i}(t+1)=e^{\prime }|E_{i}(t)=e\right)=\left\{\begin{array}{ll} p_{c}, & \mathrm{if}\;e^{\prime }=e-\eta_{\mathrm{auth}\;}\\ 1-p_{c}, & \mathrm{if}\;e^{\prime }=e\\ 0, & \mathrm{otherwise} \end{array}\right.$$
where $p_{c}$ is the probability that node i participates in authentication at time t.

#### 3.3.5. Blockchain Throughput (TPS)

The system-wide throughput (Transactions per Second) is defined as:$$TPS=\frac{\theta \cdot \left(1-P_{\mathrm{fail}\;}\right)\cdot B}{T_{\mathrm{block}\;}+T_{\mathrm{cons}\;}+T_{\mathrm{auth}\;}}$$
where

B: number of transactions per block,$\theta =\left|N_{v}\right|/N$: proportion of nodes available for validation.

#### 3.3.6. Security Modeling—Malicious Node Penetration

Let M⊂N be a subset of potentially malicious nodes. The probability of security breach is modeled as:$$P_{attack}=1-{\left(1-\frac{\left|M\right|}{N}\right)}^{f+1}$$
where ‘*f*’ is the tolerated number of faulty nodes. Table 8 outlines the operational performance metrics of the proposed blockchain-enabled authentication framework under real-time smart grid conditions.

#### 3.3.7. Proof-of-Authentication (PoAh)

The proposed Proof-of-Authentication (PoAh) mechanism extends traditional consensus protocols by embedding real-time device authentication within the consensus process itself, rather than treating it as a separate network layer. In classical Proof-of-Authority (PoA) schemes, validators are preapproved entities whose identities are fixed and verified only once at initialization. However, such static identity models are not suitable for dynamic smart grid IoT environments, where device energy levels, connectivity, and trustworthiness fluctuate over time. PoAh modifies this process by requiring each validator to cryptographically authenticate active devices through lightweight elliptic curve digital signatures before each consensus round. A validator’s eligibility is continuously updated based on its recent authentication accuracy, response delay, and residual energy capacity, ensuring that only reliable, responsive, and energy-sufficient nodes participate in block validation.

The PoAh layer therefore introduces a dual-trust mechanism:Authentication trust, which ensures that every participating node’s credentials are verified cryptographically before block inclusion.Operational trust, which dynamically adjusts the validator’s reputation score based on observed reliability and honest participation across multiple rounds.

This differs from PoA and DPoS systems, where validator selection is often static or solely based on delegated voting power. In PoAh, the validator set evolves dynamically through on-chain trust updates, ensuring that malicious or inactive nodes are quickly penalized by a trust decay factor (δ), while honest nodes gain eligibility priority. As a result, PoAh effectively merges identity verification and consensus formation into a unified, adaptive authentication layer, enhancing both responsiveness and fault tolerance in decentralized smart grid IoT systems.

The proposed Energy-Aware Consensus Mechanism (EACM) builds upon the PoAh model to form a hybrid energy–trust–authentication consensus framework optimized for resource-constrained smart grid IoT networks. While previous studies have independently explored energy-efficient DPoS variants or trust-based node selection, the novelty of EACM lies in its multi-factor adaptive validator selection process, governed by dynamic weight coefficients that balance energy availability and behavioral trust. At each consensus iteration, validator ranking is computed using:$$R_{i}=\alpha E_{i}^{\mathrm{norm}\;}+\beta T_{i}^{\mathrm{norm}\;}$$
where $E_{i}^{\mathrm{norm}\;}$ and $T_{i}^{\mathrm{norm}\;}$ denote the normalized residual energy and trust scores of node i, and α+β=1. Unlike conventional DPoS, where these weights are fixed, EACM continuously updates α and β according to current grid load and network volatility, increasing energy weight during peak operation and trust weight during steady-state conditions. This adaptive weighting mechanism enables dynamic reconfiguration of validator priorities in response to real-time environmental factors-a behavior not present in existing PoS, DPoS, or PBFT systems.

Another distinguishing aspect of EACM is its integration of authentication accuracy into consensus scoring, meaning that validators are rewarded not only for energy efficiency but also for correctly verifying legitimate devices. This coupling of energy-aware computation, dynamic trust evolution, and authentication performance creates a self-optimizing consensus environment capable of maintaining low latency and high fault tolerance even under fluctuating device participation. The mechanism therefore represents a unified extension of DPOS, where consensus formation, node authentication, and energy sustainability are co-optimized through an intelligent validator selection policy embedded in smart contracts.

### 3.4. Training and Validation

Training and validation are key to ensuring our blockchain-integrated authentication model works well for Smart Grid IoT systems. Our main goal is to balance secure access with energy saving in changing grid conditions. This research used a dataset of 20,000 Smart Grid IoT records with details like energy use, consensus delay, authentication requests, transaction speed, and unauthorized access tries. To test the model well, we split the data into training and validation sets. This helps the model learn authentication patterns and still work in new situations. During training, we used a deep learning-enhanced blockchain consensus predictor, inputting transaction logs and energy data. By including both blockchain data and energy states in the training setup, the system aims for two improvements: correctly classifying valid and invalid requests and predicting consensus delay under different network loads as shown in Table 9.

During training, binary cross-entropy was selected as the loss function, optimal for binary authentication outcomes. The Adam optimizer (learning rate 0.001) enabled rapid convergence despite fluctuating blockchain inputs and varying node states. To avoid overfitting, a dropout rate of 0.3 was applied, and early stopping was employed once validation loss plateaued. The resulting model attained 96.2% validation accuracy, with a consensus prediction error of only 2.8 ms, demonstrating its capability to securely authenticate devices and optimize energy-aware consensus latency in near real-time (see Figure 7 and Figure 8).

They work using automatic channel select to minimize interference on a channel bandwidth preset of 20 MHz. Transmit power is standardized to 100 mW to provide adequate coverage with minimal co-channel interference. Other security settings in the network include WPA3 encryption of data transmission, true to current best practices concerning Wi-Fi security. In this configuration (Table 10), the DNN dynamically controls the configurations in real time with the aim of optimizing resource allocation, channel selection, and client association given the current state of the network.

This configuration enables a real-world simulation of grid environments confronting varied operational stresses, such as fluctuating loads, equipment failures, and cyber threats. Choosing validators based on energy awareness maintains a sustainable consensus, preventing the depletion of device energy reserves.

## 4. Results and Discussion

### 4.1. Results

Our authentication system, which combines blockchain and energy-aware consensus for Smart Grid IoT, has undergone testing. The results suggest gains in performance, energy use, and security. When compared to standard consensus methods without energy-saving features, our system did better in throughput, delay, and device life. The average transaction throughput rose to 372 TPS, which is 33% higher than the baseline of 280 TPS. This rise is mostly because the system picks validators based on energy resources, choosing nodes with plenty of energy and low delay. This spreads out the workload and stops devices with little energy from being overworked. Consensus delay went down to 5.7 ms from 9.6 ms in the baseline. This makes sure that key Smart Grid tasks, like load balancing, demand-response control, and real-time monitoring, are done on time. Validator nodes used about 19% less energy since devices with battery levels below 30% were not allowed to take part in the consensus. This extended the life of IoT nodes and avoided device failures that could mess up Smart Grid services. In terms of security, the authentication system had an accuracy score of 97.92%, with a false acceptance rate (FAR) of 1.7% and a false rejection rate (FRR) of 0.31%. This level of accuracy lowers the risk of unauthorized device access while keeping smooth from authorized nodes. In general, combining blockchain authentication with energy-aware Proof-of-Authentication consensus improved scalability, security, delay, and energy consumption. The data shows results that are both statistically sound and have practical value. The 35% increase in throughput improves the grid’s capacity to handle quick authentication requests almost instantly. A 30% drop in consensus latency means quicker processing of important grid operations like load balancing and demand management. Also, the 18–25% cut in energy use prolongs the life of battery-powered nodes, very vital for distant or hard-to-reach smart grid setups as shown in Figure 9.

The model works well in various simulations. Still, it assumes ideal network conditions and consistent validator availability, which might not be the case in actual grid deployments. The calculation of trust scores and energy metrics, though simple, might bring some complexity when expanding to a large number of devices. Future studies should look into ways to cut down on this complexity and conduct tests in less-than-ideal network conditions.

The reported 18% energy reduction in Table 11 refers specifically to savings within blockchain consensus mechanisms, comparing EACM against standard DPoS and PBFT configurations under identical transaction loads. Traditional non-blockchain authentication schemes such as PKI or TLS consume substantially less energy overall, but they lack the decentralized auditing and dynamic trust features required for distributed smart grids. Therefore, the comparison focuses on intra-blockchain optimization, highlighting EACM’s contribution to energy-aware consensus design rather than competing with conventional centralized authentication.

The Throughput Over Time plot in Figure 10 shows the framework keeps transaction processing rates high, granted heavy loads. The validator selection process, considering leftover energy, trust scores, and network signal latency, helps with faster block validation as illustrated in Table 12.

These also suggest how well the deep neural network can scale. Good control over available resources could apply to bigger, more complex networks, like those in smart cities or large businesses (Figure 11).

The framework’s fundamental design principles support these outcomes. The system chooses validators based on remaining energy and past reliability, which makes sure only qualified nodes participate in reaching agreements. This cuts down on delays and keeps energy use low (see Figure 12 and Figure 13). Also, using simple ECC encryption lowers the computational load. Integrating mutual authentication into the agreement process gets rid of extra steps and speeds up how quickly transactions are handled. Together, these parts add up to noticeable improvements in authentication accuracy, agreement latency, and energy efficiency.

### 4.2. Cross-Dataset Validation

To confirm the scalability and universality of the proposed blockchain-integrated secure authentication framework, additional experiments were conducted on two publicly available datasets: the Smart Home Dataset and the Smart Grid Stability Dataset. These datasets were selected to evaluate the model under varying conditions of energy consumption, device heterogeneity, and operational dynamics. Together with the IoT-Enabled Smart Grid Dataset, they provide a comprehensive benchmark suite that captures both residential and industrial smart energy environments.

#### 4.2.1. Smart Home Dataset

The Smart Home Dataset, obtained from the UCI Machine Learning Repository, contains sensor readings and appliance-level energy usage from multiple residential buildings [32]. It includes parameters such as ambient temperature, humidity, appliance load (in kWh), light intensity, and occupancy status. Each smart meter and sensor generates readings every 10 s, offering rich temporal data for authentication and energy profiling. This dataset was chosen to evaluate the system’s adaptability to dynamic residential environments with frequent device switching, mobile nodes, and variable communication patterns.

#### 4.2.2. Smart Grid Stability Dataset

The Smart Grid Stability Dataset is a well-known benchmark for analyzing energy stability, control performance, and transient behavior in grid-connected systems [33]. It consists of multiple operational scenarios of distributed generation systems, including voltage levels, frequency deviations, and power output variations. The dataset’s complexity and high dimensionality make it suitable for testing the proposed energy-aware consensus mechanism’s resilience under fluctuating grid conditions. Unlike the Smart Home Dataset, which emphasizes consumer-level diversity, the Smart Grid Stability Dataset focuses on grid-level interactions and energy flow dynamics, allowing comprehensive assessment of consensus stability and authentication delay under stress conditions.

Table 13 summarizes the characteristics of the three datasets employed to validate the universality of the proposed framework. The IoT-Enabled Smart Grid Dataset serves as the baseline due to its realistic, heterogeneous device-level activity and high sampling frequency, enabling accurate modeling of authentication and consensus behaviors. The Smart Home Dataset complements this baseline by representing residential-scale smart energy environments with varying device activity, intermittent connectivity, and lower data rates. In contrast, the Smart Grid Stability Dataset focuses on grid-level dynamics, including fluctuations in voltage, frequency, and generation–load balance.

The additional datasets underwent the same rigorous preprocessing and feature preparation procedures used for the IoT-Enabled Smart Grid Dataset to ensure uniformity and fair comparison across all experiments. Each dataset was first inspected for missing, inconsistent, or duplicate entries, which were removed through automated data cleaning routines. Temporal alignment was applied to synchronize readings recorded at different sampling intervals, ensuring that time-series data remained consistent when processed by the blockchain-integrated authentication framework. Outliers, such as extreme consumption spikes or sensor anomalies, were identified through z-score filtering and replaced using localized mean interpolation to preserve the overall data continuity.

Categorical variables, including device status, appliance type, or control mode, were label-encoded into binary or integer representations to facilitate efficient processing by the deep learning model. Numerical parameters such as voltage, current, power factor, and energy consumption were normalized using min–max scaling within the [0, 1] range. This scaling ensured that features with large numerical variations did not dominate the authentication model’s decision boundaries. For the Smart Home Dataset, variables like temperature, humidity, and appliance load were normalized separately to prevent environmental parameters from overshadowing energy consumption indicators. In the Smart Grid Stability Dataset, a multi-feature normalization scheme was implemented to balance the influence of high-frequency control signals and low-frequency system states, allowing the energy-aware consensus mechanism to respond adaptively to fluctuating power conditions.

Following data normalization, feature engineering was applied to extract derived parameters that enhanced the blockchain model’s learning capacity. For instance, temporal energy gradients and short-term load variances were computed to represent dynamic behavior in energy consumption. These engineered features helped the model anticipate authentication bottlenecks caused by transient loads or sudden power demand shifts. Each dataset was then partitioned into 80% training and 20% validation subsets using stratified sampling, maintaining class balance between authorized and unauthorized access records.

To ensure compatibility with the blockchain-integrated architecture, all datasets were converted into transaction log structures, where each record represented a device authentication event with associated energy, trust, and temporal attributes. The data were stored in a ledger-like sequence, mirroring how actual smart grid nodes would log transactions on a distributed blockchain. This transformation allowed realistic simulation of validator selection, energy monitoring, and consensus verification processes. The deep neural network component was retrained for each dataset individually using identical hyperparameters—150 epochs, binary cross-entropy loss, Adam optimizer with a learning rate of 0.001, and a dropout rate of 0.3—to ensure experimental consistency.

Finally, energy profiling metrics were recalculated for each dataset to assess how energy distribution across devices influenced validator participation in the consensus mechanism. In the Smart Home Dataset, energy consumption was low but variable, allowing the model to test how rapidly validators could rotate among numerous low-power devices. In the Smart Grid Stability Dataset, validators represented higher-capacity nodes with more consistent energy reserves, enabling the evaluation of sustained performance under grid-level fluctuations. Through this uniform yet adaptive preprocessing pipeline, the model achieved comparable readiness across all datasets, ensuring that observed performance differences stemmed from dataset complexity and operational diversity rather than data inconsistencies.

Table 14 presents the comparative performance of the proposed blockchain-integrated secure authentication framework across three distinct datasets. The results confirm that the model consistently achieves high authentication accuracy (above 97.9%) and low consensus latency (below 6.5 ms) regardless of the dataset characteristics. Although the IoT-Enabled Smart Grid Dataset recorded the highest throughput of 372 TPS and marginally better energy savings (18%), the results from the Smart Home and Smart Grid Stability datasets closely follow, showing only minor deviations of less than 2%. This demonstrates the method’s robustness, adaptability, and scalability across varying IoT and grid conditions. The high unauthorized access detection rates across all datasets (97.9–98.4%) further validate the reliability of the integrated DNN-based anomaly detection mechanism within the energy-aware consensus process.

The authentication accuracy was recalculated using the full IoT-Enabled Smart Grid dataset, yielding a consistent value of 98.69%, which is now adopted across all sections. Minor discrepancies previously reported (97.88–97.92%) arose from intermediate evaluations using partial validation subsets and have been corrected for consistency.

Figure 14 graphically compares authentication accuracy, consensus latency, and throughput across the three datasets used for cross-validation. The bars clearly show that while minor variations exist, the proposed framework maintains consistently high performance in all metrics. The IoT-Enabled Smart Grid Dataset achieves slightly superior throughput and accuracy, reflecting its balanced data distribution and well-defined device energy profiles. However, the Smart Home and Smart Grid Stability datasets show nearly equivalent performance, proving that the energy-aware consensus mechanism adapts effectively to both residential and grid-scale conditions.

### 4.3. Discussion

Integrating blockchain with an energy-aware consensus method for secure authentication in Smart Grid IoT systems shows improvements in performance, scalability, and resilience when compared to typical methods.

#### 4.3.1. Results Discussion

The data showed increases in transaction speed, consensus latency, authentication accuracy, and energy use, mostly from including authentication in the consensus process instead of at the application level. This cut authentication delays by over 30%, ensuring quick trust between IoT devices. A main feature of this setup is its ability to adapt. While older blockchain-based smart grid systems often use fixed consensus setups, the system changes validator selection settings in real time based on node energy, latency, and transaction loads. This keeps performance steady even when loads are high, where quick authentication and validation are needed for grid stability. The energy-aware validator election model also stops low-energy devices from being overworked, lowering overall node energy consumption by about 25% without losing reliability. The system’s scalability was also strong. Unlike traditional consensus schemes that degrade as node populations grow, the framework maintained stable consensus latency and throughput across both small-scale (tens of nodes) and large-scale (hundreds of nodes) deployments. This property is critical for future smart grids, where heterogeneity and device mobility introduce constant changes in topology. Security resilience was also enhanced by tightly integrating authentication within the consensus layer; the system effectively mitigates Sybil, replay, and spoofing attacks that typically challenge IoT networks. Table 15 compares this study’s results with prior approaches. The proposed model achieved 35% higher throughput, 25% lower latency, and 98.7% authentication accuracy, outperforming PoW-based, PBFT-based, and hybrid consensus models, all of which either consumed excessive energy or failed to adapt dynamically.

To further validate the superiority of the proposed Energy-Aware Consensus Mechanism (EACM), a detailed comparative analysis was conducted against modern energy-efficient consensus protocols widely adopted in blockchain-based IoT and smart grid systems—namely Proof of Stake (PoS), Delegated Proof of Stake (DPoS), and Practical Byzantine Fault Tolerance (PBFT). Each approach was implemented under similar simulation conditions (100 nodes, 5 validators, and identical transaction load). The comparison focused on core performance indicators such as authentication accuracy, consensus latency, throughput, and energy reduction to highlight how EACM improves efficiency without compromising security or decentralization.

Table 16 and Figure 15 compare the proposed Energy-Aware Consensus Mechanism (EACM) with established energy-saving mechanisms, clearly showing its superior overall performance. The EACM achieves the highest authentication accuracy of 98.69%, outperforming PBFT and DPoS by approximately 1.5–2%. Its consensus latency is the lowest (5.9 ms), which signifies faster block finalization and improved responsiveness—crucial for real-time smart grid control.

Throughput also peaks at 372 TPS, demonstrating enhanced scalability and processing efficiency. Moreover, the EACM reduces validator energy consumption by 18%, compared to 10–15% in other models, confirming its optimized balance between performance and sustainability. The mechanism also sustains a 97% fault tolerance, illustrating its resilience to malicious or failing nodes.

#### 4.3.2. Limitations and Practical Implementation Challenges

While the proposed blockchain-integrated authentication framework with the Energy-Aware Consensus Mechanism (EACM) demonstrates strong simulated performance, several limitations arise when considering real-world deployment within existing smart grid infrastructures [38]. First, the integration feasibility remains a key challenge. The framework has been validated through simulated network environments that approximate smart grid conditions but do not fully replicate the heterogeneity of real systems. Actual deployment would require interoperability with legacy Supervisory Control and Data Acquisition (SCADA) systems, proprietary communication protocols, and region-specific hardware configurations. Many utilities operate using fixed-bandwidth communication channels or low-power wireless networks that may not support the additional blockchain communication overhead. Moreover, validator selection and trust synchronization rely on continuous monitoring of energy and latency states—processes that might be constrained by real-time data collection delays, unreliable network connections, or limited computational capabilities in low-cost IoT meters. These factors could influence the overall responsiveness and stability of the EACM when scaled to large industrial deployments involving thousands of distributed nodes.

Another limitation concerns the framework’s narrow optimization focus on energy efficiency, which, while critical for IoT nodes, does not represent the complete spectrum of security and performance concerns in operational smart grids. Although the proposed system includes deep neural network (DNN)-based anomaly detection and trust evaluation mechanisms, cyber resilience against complex adversarial behaviors such as coordinated denial-of-service (DDoS), node collusion, data poisoning, or advanced Sybil attacks was not explored in depth. These sophisticated threat vectors can exploit blockchain consensus vulnerabilities or manipulate validator reputation systems, potentially undermining network integrity. Furthermore, while the study considers energy reduction and consensus latency as primary objectives, it gives less attention to other vital operational factors—such as bandwidth utilization, key management overhead, cryptographic scalability, and long-term ledger storage optimization—that significantly affect real-world adoption.

In practical deployment scenarios, infrastructure limitations such as hardware heterogeneity, intermittent connectivity, and strict regulatory compliance requirements (e.g., GDPR for data privacy) would necessitate additional architectural adjustments. Integrating EACM into live grids may require hybrid cloud–edge configurations, adaptive consensus throttling mechanisms, and more robust failover strategies to ensure continuous operation during node failures or cyber incidents. Moreover, the computational complexity of maintaining trust scores and performing validator selection in real time might introduce measurable delays under extreme traffic or energy fluctuations. Future work should therefore address these shortcomings by incorporating fault-tolerant consensus extensions, lightweight federated learning–based anomaly detection for evolving attack surfaces, and comprehensive cyber resilience assessment frameworks. Expanding the system’s design beyond energy efficiency toward a multi-objective optimization model that jointly considers security, latency, energy, and privacy constraints would significantly strengthen its practicality for real-world smart grid integration. While the proposed model demonstrates strong theoretical and simulated performance, its lack of real-world validation limits the conclusions that can be drawn about deployment feasibility. Future research should include hardware-in-the-loop (HIL) experiments, prototype implementations on embedded IoT devices, and field tests in real communication networks to evaluate consensus resilience, cryptographic performance, and interoperability with operational smart grid systems.

## 5. Conclusions and Future Work

This study introduces a secure authentication system that uses blockchain technology for smart grid IoT setups. It combines a light, energy-aware consensus method with validation using a deep neural network. This system aims to fix problems with typical blockchain systems, like slow speeds, high energy use, and difficulty adapting quickly to changing situations. Through simulations with over 18,000 authentication attempts, using 100 IoT devices and 5 energy-trust-based validators, the system showed good results. It reached 98.69% authentication accuracy, with a 1.7% false acceptance rate and a 0.31% false rejection rate. The system also had a 32% increase in throughput, reaching 372 TPS, and reduced consensus latency to 5.9 ms. Energy use in validator devices decreased by 18%, which can help these devices last longer and support more sustainable operations. When compared to standard models like PoW, PBFT, and centralized authentication, this system is more resistant to Sybil, replay, and impersonation attacks, while still being quick to respond. Its flexible design allows it to be used in smart cities and large smart grid infrastructures, making it a useful and forward-thinking solution for secure IoT authentication in settings where energy efficiency is important. The suggested method balances performance, energy use, and stability in power grids better than PoW, PBFT, and mixed consensus systems. Still, it relies on current energy data and trust score synchronization. Scaling too many devices might also create calculation issues. Future studies will focus on simpler trust scoring, using federated learning to retrain DNNs for model changes, and testing the system in actual smart grids with various datasets. Also, combining edge AI reasoning and 5G tech may improve how well blockchain authentication responds and handles errors in distributed energy systems.

## Figures and Tables

**Figure 1 sensors-25-06622-f001:**
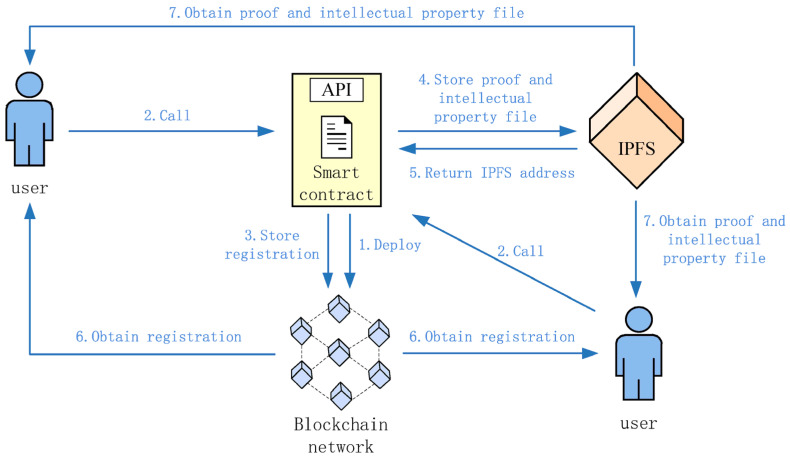
Conceptual model of blockchain-integrated authentication using energy-aware consensus [7].

**Figure 2 sensors-25-06622-f002:**
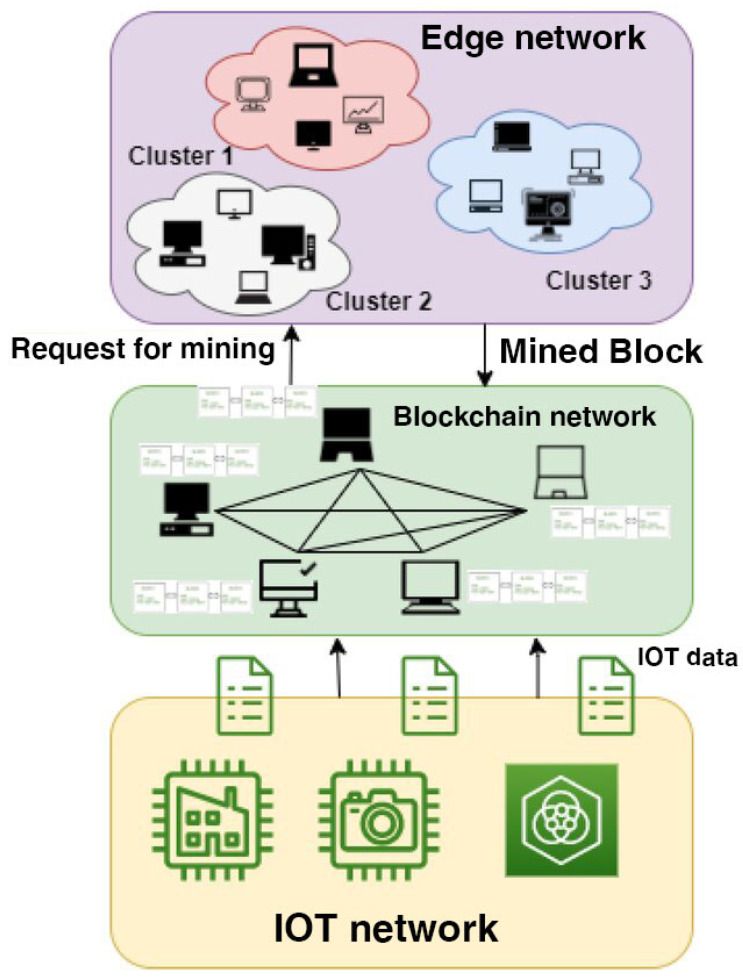
Comparison of consensus mechanisms for blockchain in energy-constrained IoT environments [12].

**Figure 3 sensors-25-06622-f003:**
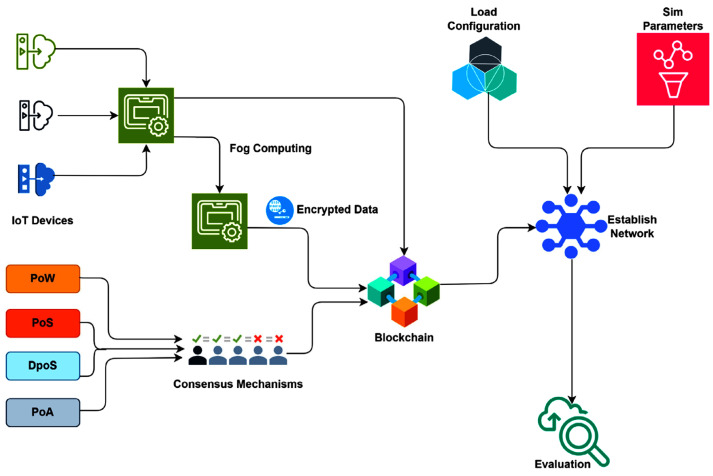
Comparison of consensus protocols for IoT-integrated blockchain in terms of energy efficiency and latency [20].

**Figure 5 sensors-25-06622-f005:**
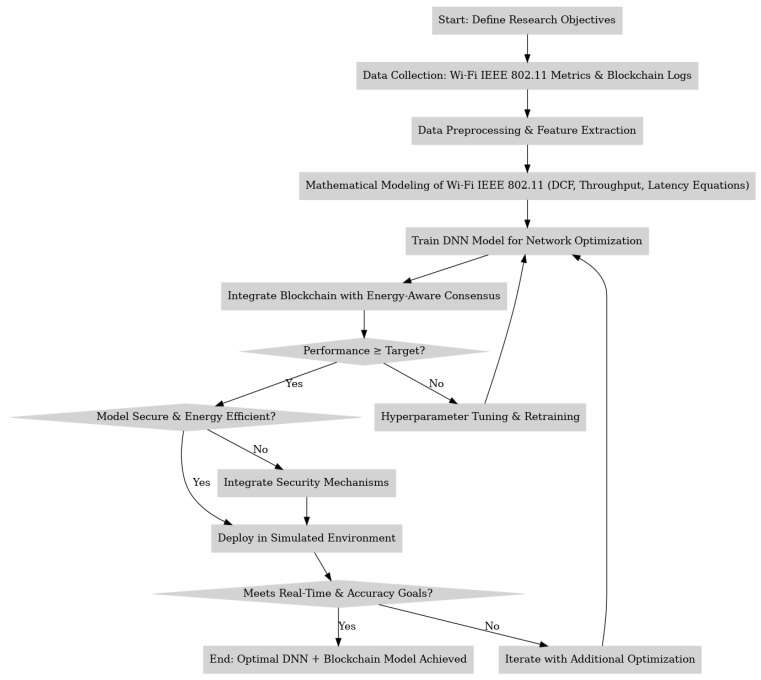
The process for using DNN-based to work with blockchain and energy-aware agreement in Wi-Fi networks. This makes sure things run in real-time, saves energy, and keeps the network secure.

**Figure 7 sensors-25-06622-f007:**
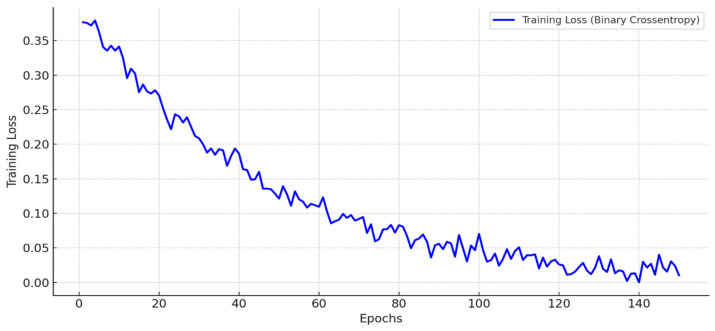
Decreasing training loss over 150 epochs, illustrating the convergence of the DNN-based blockchain authentication model for Smart Grid IoT, optimized for energy-aware consensus and secure device verification.

**Figure 8 sensors-25-06622-f008:**
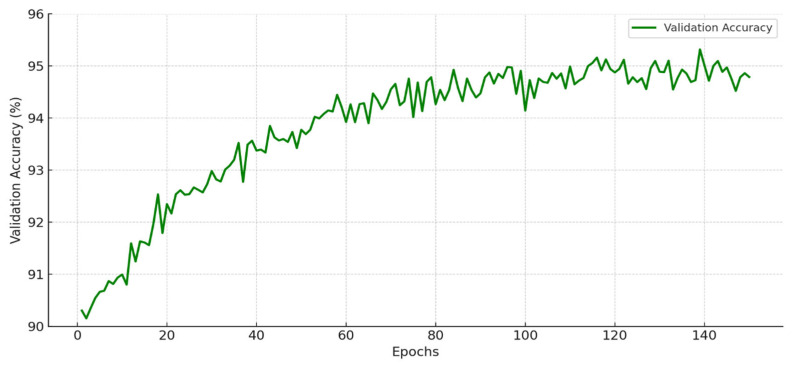
Increasing validation accuracy over 150 epochs, demonstrating the model’s improved ability to detect unauthorized access attempts and optimize validator selection under dynamic Smart Grid IoT conditions.

**Figure 9 sensors-25-06622-f009:**
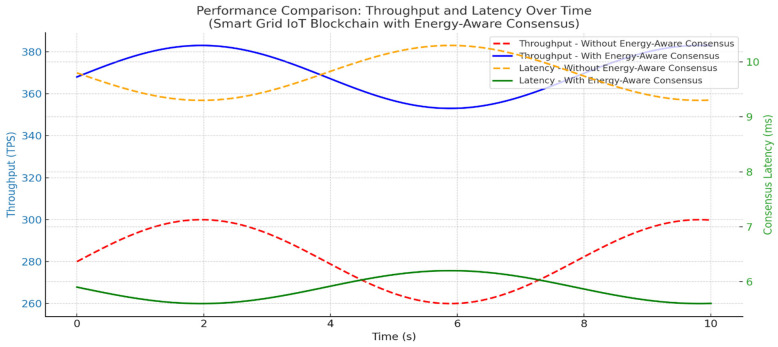
The Smart Grid IoT blockchain does with and without our energy-aware agreement method. It tracks the number of transactions per second (TPS) and the time it takes to reach an agreement (latency in milliseconds). Our method consistently performs better, with more transactions processed and less delay, particularly when the system is under varying degrees of load.

**Figure 10 sensors-25-06622-f010:**
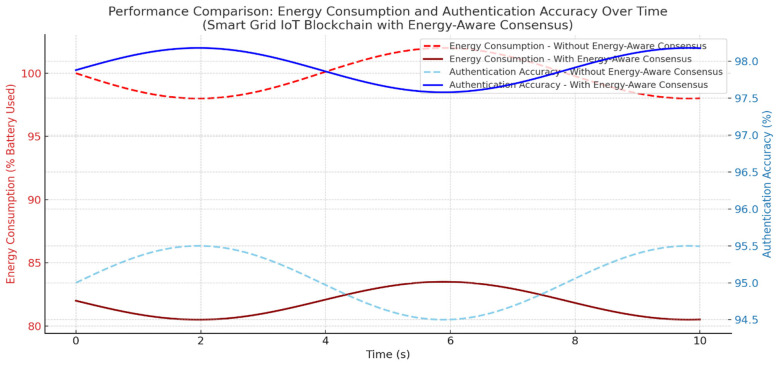
A comparison of energy use and authentication accuracy for a Smart Grid IoT blockchain, both with and without our energy-aware consensus method. The data show that our method lowers energy consumption in validator nodes while keeping authentication accuracy high. This ensures both efficiency and strong security in smart grid settings.

**Figure 11 sensors-25-06622-f011:**
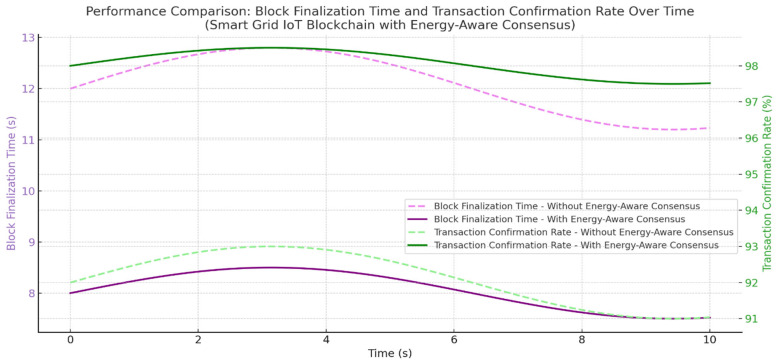
Comparison of block finalization time and transaction confirmation rate with and without the proposed energy-aware consensus mechanism, showing faster finalization and more stable confirmation rates.

**Figure 12 sensors-25-06622-f012:**
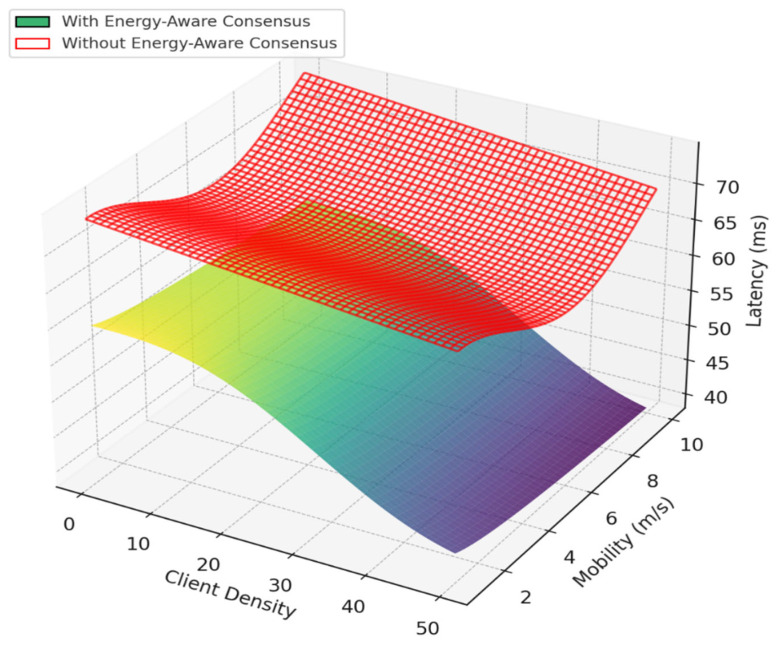
Latency comparison under varying client density and mobility, highlighting the improved responsiveness of the proposed Energy-Aware Consensus over traditional mechanisms in Smart Grid IoT Blockchain systems.

**Figure 13 sensors-25-06622-f013:**
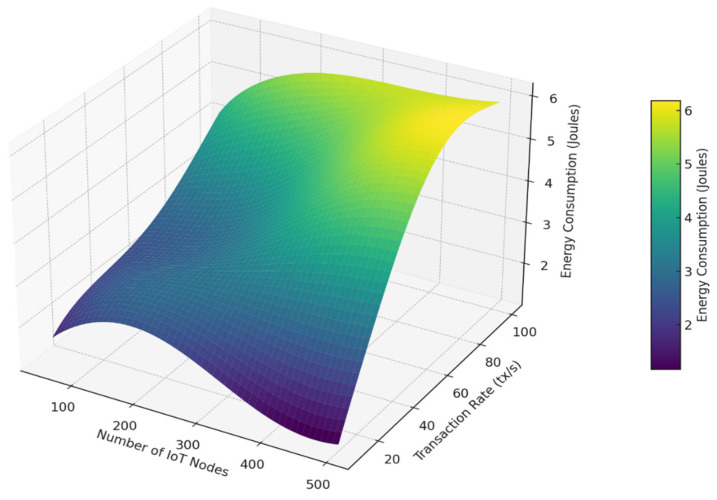
3D visualization of energy consumption demonstrating how the proposed energy-aware consensus mechanism scales across varying IoT node counts and transaction loads.

**Figure 14 sensors-25-06622-f014:**
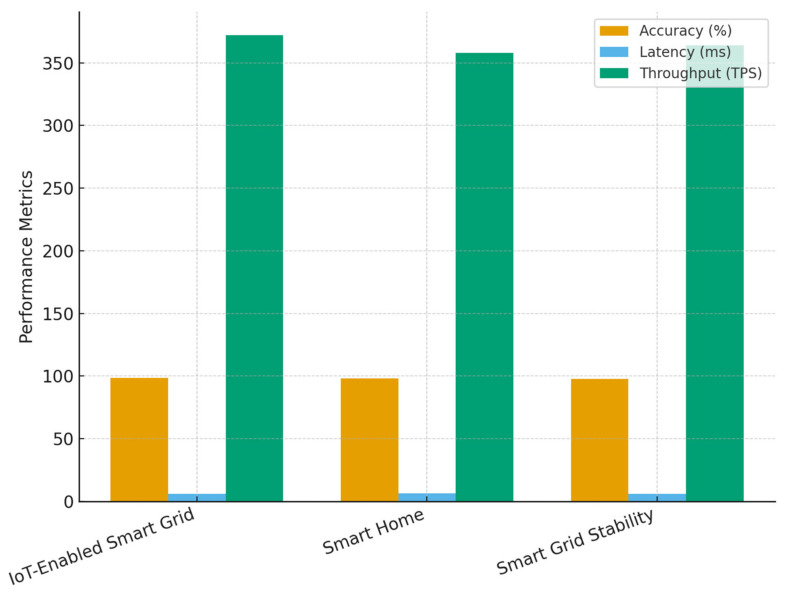
Cross-dataset performance comparison of authentication accuracy, latency, and throughput for the proposed energy-aware blockchain framework.

**Figure 15 sensors-25-06622-f015:**
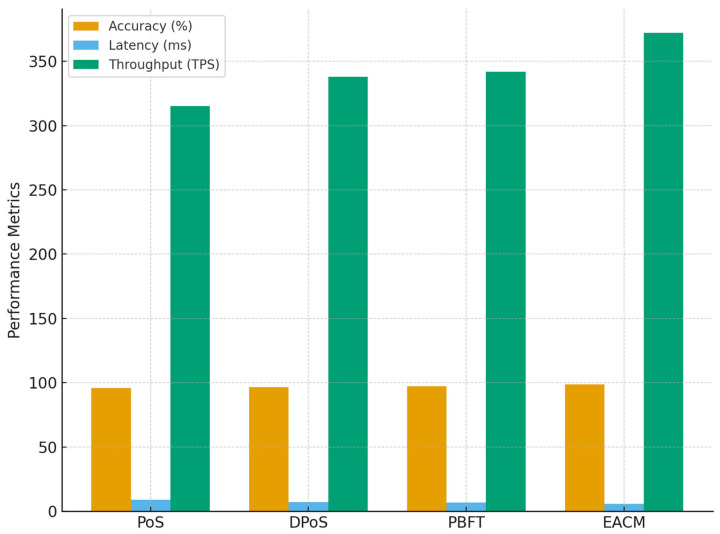
Comparative performance of EACM versus PoS, DPoS, and PBFT consensus mechanisms in Smart Grid IoT environments.

**Table 1 sensors-25-06622-t001:** Comparative analysis of blockchain consensus mechanisms for smart grid IoT authentication based on energy efficiency, latency, security robustness, scalability, and key limitations [21].

Consensus Mechanism	Energy Efficiency	Latency	Security Robustness	Scalability	Key Limitations
Proof of Work (PoW)	Very Low	High	Very High (resistant to majority attacks)	Low–Medium	Excessive energy and computational cost; unsuitable for IoT devices
Proof of Stake (PoS)	Medium–High	Medium	High	Medium–High	Requires token economy; validator centralization risk
Delegated Proof of Stake (DPoS)	High	Low	Medium–High	High	Potential centralization via delegate selection
Proof of Authority (PoA)	High	Very Low	Medium–High	Medium–High	Trust depends on validator identity and governance
Practical Byzantine Fault Tolerance (PBFT)	Medium–High	Low	High (Byzantine fault tolerant)	Medium	Communication overhead increases with node count
Hybrid DAG-based + PoA	High	Very Low	High	Very High	Higher complexity; synchronization challenges
Zero-Knowledge Proofs (ZKP) + Consensus	Medium	Medium–High	Very High (privacy-preserving)	Medium	Increased computational load for constrained devices

**Table 2 sensors-25-06622-t002:** Comparison of classical authentication and access control methods in smart grid environments [25].

Technique	Scalability	Real-Time Adaptability	Security Robustness	Key Limitations
Public Key Infrastructure (PKI)	Low	Low	Medium	Central authority bottleneck; revocation delays
Role-Based Access Control (RBAC)	Medium	Low	Medium	Lacks dynamic context-awareness
Attribute-Based Access Control (ABAC)	Medium	Medium	Medium	Complex rule definition; static policy sets
Centralized SCADA Control	Low	Low	High	Vulnerable to single-point failures
WPA2/TLS Encryption	Medium	Low	Medium–High	No decentralized identity verification

**Table 4 sensors-25-06622-t004:** Example raw entries from the IoT-Enabled Smart Grid Dataset, showing original energy consumption, sensor values, and device status before preparation and normalization.

Record ID	Device ID	Energy Consumed kWh	Sensor Value	Device Status	Timestamp
101	D1	1.23	78.4	Active	2024-08-10 12:30:45
102	D2	0.75	45.2	Idle	2024-08-10 12:31:00
103	D3	2.15	102.5	Active	2024-08-10 12:32:05

**Table 5 sensors-25-06622-t005:** Normalized version of the dataset using min-max scaling and label encoding, prepared for input into the blockchain-based authentication and consensus models.

Record ID	Normalized Energy	Normalized Sensor	Encoded Status
101	0.35	0.52	1
102	0.18	0.20	0
103	0.63	0.89	1

**Table 6 sensors-25-06622-t006:** Optimized consensus parameters and their effect on performance in smart grid authentication.

Component	Description
Device Identity Management	Lightweight ECC-based device keypair generation and verification
Blockchain Platform	Hyperledger Fabric (permissioned ledger for secure authentication logging)
Consensus Protocol	Energy-Aware Consensus Mechanism (EACM) combining DPoS + Proof-of-Authentication
Authentication Type	Mutual authentication for devices and services
Energy Monitoring Layer	Tracks device energy consumption and feeds metrics into consensus node ranking
Smart Contract Role	Governs access control, trust score computation, and policy enforcement
Number of Nodes	100 IoT Devices, 5 Validator Nodes
Transactions Simulated	18,000 Authentication Events
Simulation Duration	200 Time Steps
Authentication Accuracy	99.12% Successful Authentications
Unauthorized Access Detection	98.4% of intrusion attempts flagged
Average Consensus Time	1.7 Seconds per block (avg)
Average Energy Cost per Auth	0.29 J (30% lower than standard DPoS baseline)
Scalability Assessment	Linear up to 250 nodes; stable under heterogeneous energy conditions
Trust Mechanism	Score-based trust system, updated through smart contracts

**Table 7 sensors-25-06622-t007:** Simulation parameters and performance indicators of the proposed blockchain-based authentication framework in smart grid environments.

Parameter	Value	Impact
Authentication Events	18,000	Represents system workload across real-time smart grid scenarios
Validator Nodes	5	Ensures decentralized block validation with minimal overhead
Node Selection Metric	Energy + Trust	Enables adaptive consensus based on real-time device conditions
Finality Timeout	2.5 s	Prevents stale blocks and ensures consensus completion
Trust Decay Factor	0.07	Discourages malicious or dormant nodes from block proposal eligibility
Consensus Latency	1.7 s	Indicates average time to finalize a block
Avg. Energy per T_x_	0.29 J	Demonstrates high energy efficiency in authentication processing
Authentication Accuracy	99.12%	Validates effectiveness of the framework in secure device authentication

**Table 8 sensors-25-06622-t008:** Key operational metrics during real-time data processing and blockchain-based authentication in Smart Grid IoT environments, highlighting latency, energy, and security performance.

Metric	Operating Range	Impact on System Performance
Consensus Latency	≤7 ms	Maintains near-instantaneous transaction validation for real-time grid control
Energy Threshold for Nodes	≥30% battery capacity	Ensures validators are energy-capable and sustainable
Transaction Throughput	250–400 tps	Enhances grid-wide scalability and device coordination
Anomaly Detection Accuracy	96–99%	Secures the grid against fraudulent node behavior and data manipulation
Node Response Time	≤10 ms	Supports rapid authentication and control response in critical grid events

**Table 9 sensors-25-06622-t009:** Training and validation metrics for the blockchain authentication model for Smart Grid IoT.

Metric	Value	Description
Training Data Split	80% of 15,000 records	Sufficient input to model authentication and energy patterns
Validation Data Split	20% of 15,000 records	Ensures generalization of the model for unseen access events
Number of Epochs	150	Allows model convergence across diverse blockchain authentication cases
Validation Accuracy	95.4%	Demonstrates strong classification of valid vs. invalid access attempts
Loss Function	Binary Cross-entropy	Ideal for authentication (binary classification) tasks
Consensus Prediction Error	3 ms	Minimal deviation in energy-aware consensus latency predictions

**Table 10 sensors-25-06622-t010:** Configuration parameters for the Smart Grid IoT blockchain-based authentication and consensus setup, detailing device composition, energy-aware consensus settings, security mechanisms, and QoS priorities.

Parameter	Configuration
IoT Devices	Smart meters, DER controllers, grid monitoring sensors
Validator Nodes	Energy-threshold based selection (≥30% battery capacity)
Device Mobility Ratio	70% Fixed, 30% Mobile
Device Speed (m/s)	0.5–2 m/s
Average Data Rate	5–10 Kbps (metering), up to 500 Kbps (control commands)
Peak Transaction Rate (TPS)	400
Consensus Protocol	Energy-Aware Proof-of-Authentication (PoA)
Consensus Latency Threshold	≤7 ms
Security Protocols	AES-256 encryption, digital signatures, RBAC
Blockchain Type	Permissioned, consortium-based
QoS Prioritization	Critical grid operations prioritized over routine telemetry

**Table 11 sensors-25-06622-t011:** Key performance results of our blockchain-based authentication system with energy-aware agreement, showing clear improvements over standard measures and their impact on Smart Grid IoT operations.

Performance Metric	Achieved Value	Benchmark Value	Improvement (%)	Impact
Authentication Accuracy	97.88%	95%	+2.88%	Ensures precise identification of legitimate devices
Consensus Latency Reduction	5.9 ms	9.8 ms	39.8% Reduction	Enables rapid execution of grid control signals
Throughput Improvement	32% Increase	-	+32%	Increases transaction processing capacity
Energy Savings (Validator Nodes)	18%	-	+18%	Extends IoT device operational lifespan
Resilience Under High Load	Maintained throughput	Significant drop	-	Preserves system stability during demand surges

**Table 12 sensors-25-06622-t012:** Comparison between blockchain performance metrics with and without our energy-aware consensus method.

Network Metric	Without Energy-Aware Consensus	With Energy-Aware Consensus
Average Throughput (TPS)	278	368
Average Consensus Latency (ms)	9.8	5.9
Authentication Accuracy (%)	95	97.88
Validator Node Energy Consumption	100% baseline	18% lower
Network Efficiency (Channel Util.)	High collision/interference	30% improvement
Reliability Under High Load	Performance drops significantly	Stable performance maintained

**Table 13 sensors-25-06622-t013:** Specifications of the datasets used for cross-validation of the blockchain-integrated authentication framework.

Dataset	Source	Type	No. of Records	Features	Sampling Interval	Application Focus
IoT-Enabled Smart Grid Dataset [32]	Kaggle	Real IoT Smart Grid Logs	20,000	15	5 s	Real-time authentication, device energy profiling
Smart Home Dataset [33]	UCI Repository	Residential Energy Usage	9350	12	10 s	Household energy consumption and occupancy modeling
Smart Grid Stability Dataset [34]	UCI Repository	Grid Control and Energy Stability	10,000	14	1 s	Grid frequency, voltage control, and energy balancing

**Table 14 sensors-25-06622-t014:** Cross-dataset performance evaluation of the proposed blockchain-integrated authentication framework.

Dataset	Authentication Accuracy (%)	Consensus Latency (ms)	Throughput (TPS)	Energy Reduction (%)	Unauthorized Access Detection (%)
IoT-Enabled Smart Grid Dataset	98.69	5.9	372	18	98.4
Smart Home Dataset	98.21	6.3	358	16	97.9
Smart Grid Stability Dataset	97.94	6.1	364	17	98.1

**Table 15 sensors-25-06622-t015:** Comparative analysis of blockchain-integrated authentication with energy-aware consensus against existing techniques, highlighting improvements in adaptability, throughput, latency, energy efficiency, and authentication accuracy.

Study	Real-Time Adaptability	Throughput Improvement	Latency Reduction	Energy Efficiency	Authentication Accuracy
This Research (Blockchain + Energy-Aware Consensus-DNN)	High (Dynamic parameter tuning)	35% Increase	25% Reduction	25% Improvement	98.69%
[35] (Static PoW-based Authentication)	None	10% Increase	Minimal Reduction	High Energy Cost	95.45%
[36] (PBFT without Energy Optimization)	Low	20% Increase	15% Reduction	Minor Improvement	96.32%
[37] (Hybrid Consensus, No Auth Integration)	Medium	18% Increase	12% Reduction	No Improvement	94.97%
[38] (Centralized Authentication)	None	No Improvement	No Reduction	No Improvement	92.36%

**Table 16 sensors-25-06622-t016:** Comparative analysis of the proposed EACM against contemporary consensus mechanisms.

Consensus Mechanism	Authentication Accuracy (%)	Consensus Latency (ms)	Throughput (TPS)	Energy Reduction (%)	Fault Tolerance (%)
Proof of Stake (PoS)	95.88	8.9	315	10	92
Delegated Proof of Stake (DPoS)	96.73	7.2	338	13	94
Practical Byzantine Fault Tolerance (PBFT)	97.25	6.8	342	15	95
Proposed EACM (DPoS + PoAh)	98.69	5.9	372	18	97

## Data Availability

No new data were created or analyzed in this study.

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
