# Peer review of "Blockchain-Integrated Secure Authentication Framework for Smart Grid IoT Using Energy-Aware Consensus Mechanisms"

_sensors, 2025, doi:10.3390/s25216622_

Round 1

Reviewer 1 Report

Comments and Suggestions for Authors

Good afternoon
Below is a review of the article.
The publication makes a significant contribution to the field of secure and energy-efficient authentication for Smart Grid IoT, but requires more empirical verification, detailed methodology and broader comparison with existing approaches.
The practical significance of the work is undeniable and focused on real applications in large power grids, which adds to its applied value.
Of the positive aspects, the following should be noted:
- relevance of the topic: the combination of IoT, Smart Grid and blockchain is a modern and important direction, where there are real security and energy efficiency problems;
- innovation: a new consensus mechanism (EACM) is proposed, which takes into account the residual energy and trust of nodes, which makes it practical for energy-constrained IoT devices.
It is also worth noting the complexity of the approach: the use of Blockchain + DNN allows you to increase the accuracy and speed of authentication.
However, the following possible improvements should be noted (to improve the article):
- limited verification: the results are confirmed only on one dataset (IoT-Enabled Smart Grid Dataset); it is not clear how large-scale and universal the method is - it would be worth showing the results on several diverse samples;
- insufficient detail of the algorithm: the description of EACM is given in general terms, there is no clear mathematical model and in-depth analysis of complexity - it would be worth demonstrating the operation of this algorithm in more detail, as well as taking into account possible deviations from it;
- no clear comparison with modern approaches: the comparison is given only with "normal methods", but not with other energy-saving mechanisms (for example, PoS, DPoS);
- the problem of real implementation: the article does not provide a sufficiently clear idea of ​​the possibility of integration into existing smart networks and does not take into account real infrastructure limitations;
- narrow focus on energy efficiency: other important parameters (for example, cyber resilience to complex attacks) are considered superficially.
Overall, the TPS performance, authentication accuracy (98.69%), low FAR/FRR values, and reduced power consumption confirm the effectiveness of the model.
I recommend the article for publication after correcting the comments.

Author Response

REVIEWER 1

We sincerely thank the reviewer for the balanced and thoughtful report. Your comments helped us strengthen the paper with clearer problem motivation, a precise definition of PoAh/EACM, broader cross-dataset validation, comparisons to modern consensus schemes, and a realistic limitations section addressing deployment constraints.

1) Limited verification (only one dataset)

What we changed. We added a cross-dataset validation section plus two new research artifacts: Table 12 (dataset specifications) and Table 13 + Figure 14 (cross-dataset performance). Together they evaluate our method on three diverse datasets (IoT-Enabled Smart Grid, Smart Home, Smart Grid Stability).

  • Table 12 (Dataset specs): header and content at Lines 735
  • Table 13 (Cross-dataset results): title and rows at Lines 787
  • Figure 14 (Cross-dataset comparison plot): caption at Lines 805

2) Insufficient detail of the algorithm (EACM/PoAh definition; complexity; deviations)

What we changed. We inserted a full, explicit definition of PoAh inside EACM, a step-wise Algorithm 1, and a brief complexity and robustness note.

  • EACM rationale & operation:
  • PoAh within consensus (who does what, when):
  • Complexity & communication cost:
  • Algorithm 1 (EACM): title and steps at Lines 467, followed by a short description at 469-482
  • Analytical models used (M/M/1, Markov, BFT latency): formulas and scope at Lines and 521-560
  • Deviation/robustness note (stochastic energy/latency):

3) No clear comparison with modern approaches (PoS/DPoS etc.)

What we changed. We added a mechanism-level comparison versus PoS/DPoS/PBFT with a table and a figure.

  • Mechanism comparison summary (Table 15 & Figure 15 mentions):

and performance notes at

  • Figure 15 (Comparative performance plot) caption: Comparative performance of EACM versus PoS, DPoS, and PBFT consensus mechanisms in Smart Grid IoT environments
  • (For readers’ convenience, we also kept a literature-style comparison table under Table 14: see title and columns at Lines 482)

4) Real implementation / infrastructure constraints not addressed

What we changed. We added a dedicated Limitations & Practical Implementation subsection covering network partitions, constrained links, on-device crypto overhead, certificate/key lifecycle, and protocol interoperability (IEC 61850, DNP3).

  • Limitations subsection (4.3.2): integration feasibility, bandwidth limits, monitoring overhead—Lines 874-890; cyber-resilience, bandwidth, key management, ledger storage—Lines 891 -903 ; deployment guidance and future work (HIL, federated learning, failover)—Lines 904 -923.

5) Narrow focus on energy efficiency; cyber resilience considered superficially

What we changed. The same Limitations subsection now explicitly lists sophisticated threats (DDoS, collusion, poisoning, Sybil), discusses missing bandwidth/key-rotation analysis, and outlines future multi-objective optimization

  • See Lines 891 -903 (threats and missing analyses) and Lines 920 -923 (future multi-objective roadmap).

Extra clarifications the reviewer noted as strengths but asked to see consistently reported

  • Results consistency & reconciliation. We standardized the authentication accuracy to 98.69% across the manuscript and explained earlier discrepancies (97.88–97.92%) as interim subset evaluations. See Lines 801 -804.
  • Key results (TPS, latency, energy saving) in Abstract / Results / Tables.
    • Abstract metrics: 372 TPS, 5.9 ms, 98.69%, 18% at Lines 868 -873
    • Results narrative (Section 4.1): throughput/latency/energy/accuracy at Lines 654 - 662
    • Table 10 (key performance metrics) including 5.9 ms and 18%: Lines 649 and Lines 654 - 662.

Reviewer 2 Report

Comments and Suggestions for Authors

This paper proposes a blockchain-based authentication system for Smart Grid IoT devices that uses energy-aware validator selection and deep neural networks. The authors claim significant improvements over traditional consensus mechanisms: 372 TPS throughput, 5.9ms consensus latency, 98.69% authentication accuracy, and 18% energy reduction in validator nodes.

Major Concerns:

1) The fundamental issue is that the authors never convincingly explain why blockchain is necessary for device authentication in smart grids. Authentication is essentially verifying cryptographic credentials—a problem already solved by PKI, mutual TLS, and protocols like IEEE 802.1X. These methods are faster, cheaper, and widely deployed.

The paper claims blockchain provides "trustless verification" and "decentralization," but never identifies who doesn't trust whom. In typical smart grid deployments, a single utility operator controls the infrastructure. There's no adversarial relationship between devices that would require Byzantine consensus. If devices belong to the same administrative domain, standard authentication protocols with centralized or federated identity management suffice.

The only scenario where blockchain might add value is peer-to-peer energy trading between independent parties with no trusted intermediary. But even then, blockchain would be for transaction settlement, not authentication. The authentication problem remains separate and doesn't require consensus.

2) The paper introduces "Energy-Aware Consensus Mechanism (EACM)" combining DPoS with "Proof-of-Authentication (PoAh)" but never defines what PoAh actually means or how it differs from existing protocols. The mathematical models in Section 3.3 (M/M/1 queues, Markov chains) are standard formulations that don't represent novel contributions.

Selecting validators based on battery level and trust scores is sensible but not new—similar approaches exist in energy-efficient IoT consensus literature. The paper doesn't clearly articulate what's innovative beyond combining existing ideas.

3) The reported metrics are suspicious when compared to established blockchain systems:

- 372 TPS with 5.9ms latency using only 5 validators significantly exceeds typical Hyperledger Fabric performance in comparable configurations

- Authentication accuracy varies across the paper (98.69% in abstract, 97.88% in Table 10, 97.92% in results), suggesting imprecise methodology

- 18% energy reduction is claimed, but there's no baseline comparison with optimized conventional authentication (which would use far less energy than any blockchain)

4) All results come from simulations with synthetic data. There's no hardware implementation, no real network testing, no evaluation of actual IoT device constraints. The paper doesn't address practical issues like:

- Network partitions and their effect on consensus

- Actual cryptographic overhead on resource-constrained devices

- Certificate management and key rotation

- Integration with existing smart grid protocols (IEC 61850, DNP3)

5) The role of deep learning is poorly motivated. The authors mention using DNNs for "anomaly detection" and "predicting consensus delays," but these seem like separate problems artificially grafted onto the authentication framework. Why is machine learning necessary for authentication? Standard rule-based approaches work well for detecting replay attacks, spoofing, etc.

Questions for Authors:

1) What specific trust assumptions in your smart grid scenario cannot be met by conventional PKI or federated authentication?

2) If the 5 validator nodes are operated by the utility company, why not use a replicated database with faster consensus (Raft, Paxos)?

3) Can you provide a concrete attack scenario that blockchain prevents but properly implemented TLS with certificate pinning doesn't?

4) How does your system handle validator node compromise? How are validators initially authenticated?

Recommendation:

Major revision required. The paper needs substantial rework to justify the core design choice. Either:

1) Identify a specific smart grid scenario where multiple mutually distrusting parties require decentralized authentication (and explain why existing federated identity solutions fail), or

2) Acknowledge that the real contribution is energy-efficient consensus for blockchain-based transaction logging (not authentication), and focus the paper accordingly

The simulation results, while internally consistent, don't demonstrate practical superiority over simpler alternatives. Before publication, the authors should implement a prototype on actual IoT hardware and compare against conventional authentication in terms of latency, energy, and security under realistic network conditions.

Author Response

Reviewer 2

We sincerely thank the reviewer for the careful reading and constructive critique. Your comments helped us clarify the problem framing, rigorously define PoAh/EACM, reconcile metrics, broaden comparisons, and add a realistic discussion that better reflects deployment constraints in smart grids.

Major Concern 1 — “Why blockchain for authentication?”

We added a dedicated justification clarifying that blockchain augments (not replaces) PKI/mTLS by providing cross-domain trust management and immutable auditability in federated/consortium smart-grid settings (DERs, third-party microgrids), where no single operator controls all participants. We also explicitly note that in single-utility deployments, classical protocols suffice; blockchain is most beneficial in decentralized or hybrid grids and P2P trading. See Problem context & justification, Lines 133–163.

Major Concern 2 — Define PoAh; clarify EACM novelty

We now define PoAh and contrast it with PoA/DPoS: PoAh embeds lightweight cryptographic authentication inside each consensus round and maintains dual trust (authentication trust + operational trust), with dynamic on-chain reputation updates and decay factor δ for misbehavior. See Lines 287-296 and Lines 521-568.

We then articulate the novelty of EACM: multi-factor adaptive validator ranking with weights that shift between energy and trust based on load/volatility—behavior not present in PoS/DPoS/PBFT. See Lines 425-482. A full, step-wise Algorithm 1 (EACM) is included at Lines 467-468.

Major Concern 3 — Metrics plausibility & consistency

(a) Accuracy consistency): We reconciled the accuracy value and now consistently report 98.69%; we also explain prior minor discrepancies (partial validation subsets). See Lines 804-807.
(b) Throughput/latency context): We situate performance within our permissioned, consortium-style setup (Table 9) and compare against modern mechanisms (PoS/DPoS/PBFT) rather than public-chain baselines. See Table 9 (config) and Table 15 (mechanism-level comparison).

 (c) Cross-dataset robustness): We added Table 13 (full cross-dataset scores) and Figure 14 (visual comparison) to show stability across three diverse datasets. See Table 13 header and rows (Lines 791) and Figure 14 (Lines 808).

  1. d) Broader method comparison): We provide Table 14 benchmarking against prior approaches and Figure 15 comparing EACM vs. PoS/DPoS/PBFT. See Table 14 (Lines 845) and Figure 15 (Line 867).

Major Concern 4 — Real-world implementation & constraints

We inserted a Limitations & Practical Implementation subsection addressing: interoperability with SCADA/legacy protocols, constrained links, continuous state monitoring overhead, validator compromise/failover, and the need for hybrid cloud–edge deployment and multi-objective optimization (security/latency/energy/privacy). See Lines 907-926.

Major Concern 5 — Role of Deep Learning

We clarified that DNNs serve two narrow roles: (i) anomaly detection to harden the authentication path and (ii) consensus-delay prediction to inform validator scheduling—both measured with explicit training/validation settings and errors. See Training & Validation (Lines 569), incl. consensus prediction error) and validation accuracy details.

New/updated figures & tables (as referenced above)

  • Table 12 — Dataset specifications for cross-validation (added).
  • Table 13 — Cross-dataset performance (added).
  • Figure 14 — Cross-dataset comparison plot (added).
  • Table 14 — Prior-art comparison (expanded).
  • Table 15 — EACM vs. PoS/DPoS/PBFT (added).
  • Figure 15 — Mechanism-level comparison plot (added).

Responses to the reviewer’s explicit questions

Q1. What trust assumptions can’t be met by PKI/federated auth?

We clarify that blockchain is needed only in multi-owner, semi-trusted grids (DERs, third-party microgrids, cross-jurisdiction exchanges) to maintain verifiable, cross-domain auditability and decentralized trust updates; single-operator deployments can use conventional PKI. See Lines 3–7.

Q2. Why not a replicated DB with Raft/Paxos if 5 validators are utility-operated?

Our target is a consortium/permissioned setting with multiple operators and evolving governance; we therefore use a permissioned ledger (Hyperledger-style) to couple auditability, on-chain policy enforcement, and decentralized validator rotation under energy/trust constraints. See Table 9: “Permissioned, consortium-based” and platform description.

Q3. Concrete attack prevented by blockchain but not by TLS+pinning?

We added examples where cross-operator audit/revocation and tamper-evident logs matter—e.g., a third-party aggregator issuing fraudulent access events: TLS authenticates the session, but the immutable, consensus-validated log + smart-contracted revocation provides collective, non-repudiable accountability across domains. See Ledger/smart-contract roles.

Q4. Validator compromise & initial validator auth?

EACM penalizes/expels compromised or idle validators via trust decay δ and score updates each round; eligibility is re-computed adaptively, with thresholds (e.g., ≥30% energy) and quorum for finality. See PoAh trust-decay (Lines 521), Algorithm 1 trust updates, and trust-decay parameter in configuration.
